# Detection and Prevention of DDoS Attacks on the IoT

Shu-Hung Lee [1], Yeong-Long Shiue [2], Chia-Hsin Cheng [2,*], Yi-Hong Li [2] and Yung-Fa Huang [3,*]

1   School of Intelligent Manufacturing and Automotive Engineering, Guangdong Business and Technology University, Guangdong 526020, China
2   Department of Electrical Engineering, National Formosa University, Yunlin 632301, Taiwan
3   Department of Information and Communication Engineering, Chaoyang University of Technology, Taichung 413310, Taiwan
*   Correspondence: chcheng@nfu.edu.tw (C.-H.C.); yfahuang@cyut.edu.tw (Y.-F.H.); Tel.: +886-4-2332-3000 (Y.-F.H.)

**Abstract:** The Internet of Things (IoT) system has been a hot topic in recent years. Its operation is a system that stores data in data storage and is completed by the exchange of network information about things. Therefore, the security of information between network transmissions is very important. In recent years, the most likely cause of information security problems has been a distributed denial of service (DDoS) attack. In this paper, we proposed an autonomous defense system that combines edge computing with a two-dimensional convolutional neural network (CNN) to recognize whether the data server in IoT suffers from DDoS attacks and identify the attack mode. The accuracy of trained two-dimensional CNN is up to 99.5% and 99.8% for packet traffic and packet features training, respectively. A field experiment's results show that the data server in the proposed system can effectively distinguish the difference between the DDoS attacks and the normal transmission to reduce the impact of DDoS attacks on the IoT data storage while it is under attack.

**Keywords:** Internet of Things; distributed denial of service; convolutional neural network; edge computing

## 1. Introduction

In recent years, the Internet of Things (IoT) [1] based on Wireless Sensor Networks (WSNs) has become more and more developed. The purpose of the IoT is to access data from each other through the remote connection of machine to machine (M2M) without human operation. The architecture of the IoT can be divided into three layers, named the sensing layer, network layer, and application layer. To meet the heterogeneous network architecture in the IoT system, most people choose to add a network layer gateway for heterogeneous network processing and upload the processed information to the cloud. Most gateways only process the heterogeneous network information and configure sufficient minimum storage capacity. The limitation of storage capacity makes it impossible for users to install anti-virus software on IoT devices, resulting in many potential loopholes.

With the rapid development of network technology, the speed of data exchange between devices is getting faster and faster. In the case of vulnerabilities in the system software or firmware, attackers can not only steal the data collected from the device for selling personal data, phishing, spreading spam, etc., but can even launch distributed denial of service (DDoS) attacks against other targets by controlling the device [2]. It is difficult for common users to detect the device being attacked or controlled for the first time until network resources or system resources are affected and network services are temporarily delayed or interrupted. Therefore, we hope to make future IoT devices more secure by analyzing network packets to prevent blocking attacks.

There is already much literature proposing a variety of methods to lessen the effect of DDoS on IoT [3–6]. In addition, with the development of artificial intelligence, various algorithms have been applied in different fields. Deep Learning is a framework belonging

to artificial intelligence, which is derived from the Neural Network (NN) framework improved by Hinton et al. in 2006 [7]. It can analyze sequential data and has been applied to many fields, such as speech recognition, image recognition, natural language processing, etc. Through [8–12], it can be demonstrated that deep learning can be applied to intrusion detection systems and can effectively improve the efficiency of network information security detection.

However, the identification results on the edge computer when measured under different conditions, in which Feature is the identification result of the feature model, and Flow means the identification result of the traffic model. Therefore, we conducted our research using the related neural network approach. The proposed architecture of edge computing with a trained CNN model will make correct identifications under the normal transmission, SYN flood attack, UDP flood attack, ICMP flood attack, and MIX flood attack, respectively. In this proposal, we would like to maximize the use of information about changes in related characteristics when packets are transmitted in the system, which can effectively improve the accuracy of recognition and make up for the shortcomings of using a single model for judgment.

Some relevant pieces of literature about machine learning (ML) based methods to mitigate DDoS attacks in IoT systems in recent years will be reviewed below. An evaluation of using a Random Neural Network (RNN) trained by normal traffic data compared to the Long-Short Term Memory (LSTM) to detect SYN flood DDoS attacks was provided in [13]. It shows better results in RNN than LSTM, but its accuracy was almost 81% and not considered better enough to rely on. In [14], a new deep neural network to identify the network flows being normal or abnormal was presented. The authors adopted a feedforward back-propagation design with seven secret layers and tested the method for DDoS detection using the most up-to-date Canadian dataset (CIC IDS 2017). The test provided a value of 0.99 scores which means that the experimental results were accurate in terms of Recall and Precision. A resource-friendly ML algorithm called Edge2Guard (E2G) was introduced in [15]. It was trained and tested over the N-BaIoT dataset of normal and attack network traffic logs recorded by using Mirai and Bashlitte Botnets. The algorithm has resource-friendly detection depending on creating an E2G model for each MCU-based IoT device separately in the system. The disadvantage of this algorithm is that the model should be upgraded frequently after being trained with data from the developed type of malware action resulting in rising difficulties in the deployment process. An effective method employing two essential attributes named Volumetric and Asymmetric to detect two forms of flooding-based DDoS attacks was presented in [16]. The proposed DDoS attack detection method based on SDN can cause minimal disruption to effective user activity and reduce both training and testing time.

In addition, it proposed the Advanced Support Vector Machine (ASVM) technique to enhance the current Support Vector Machine (SVM) algorithm to detect DDoS flooding attacks effectively. In [17], a new detection classification system based on SVM and CNN ML algorithms was proposed. It converts the binary files into visualized images in grayscale. Then the CNN and SVM process these images to detect if a file contains maliciously injected code. The accuracy of this method is up to 94% in the binary classification case but only 81% in the multi-classification case. A new ML method based on clustering and graph structure features to predict the occurrence of DDoS attacks was provided in [18]. The method creates the edge and vertex structures in graph theory and extracts eight features of traffic data as input variables. Then uses the principal component analysis (PCA) model to extract the features of DDoS and normal communication. Finally, the fuzzy C-means (FCM) clustering method is used to detect DDoS. The availability of this method is verified by taking 2000 traffic data in CICIDS-2017 as an example. The results of recall, false positive, true positive, true negative, and false negative were 100.00%, 1.05%, 68.95%, 0.00%, and 30.00%, respectively, indicating that it improves the reliability of detection and has a good detection effect on DDoS attacks compared to other methods.

In the IoT system, due to the different wireless sensing networks used by the sensing layer, most of them need a gateway for cross-heterogeneous network processing. However, most gateways are used for heterogeneous network processing and do not have a lot of storage space. As a result, anti-virus software cannot be installed on the device, resulting in security vulnerabilities in the system. That makes the storage devices of the IoT the target of malicious attacks tested by novice hackers, among which the most common malicious attack mode includes DDoS attacks, which exhaust the target network or system resources through flood attacks, thereby slowing down or terminating their services. How to detect malicious attacks will be the focus of this paper.

In the past, the method of preventing DDoS attacks was to track the source of the attack and block the attack through functions such as Intrusion Detection Systems (IDS) and firewalls. However, with the increasing scale of DDoS attacks in recent years, such as the Mirai zombie virus in 2016, the Dyn.com domain name systems (DNS) services company, Domain Name Services suffered a DDoS attack, affecting websites such as Cable News Network and Twitter [19]. This kind of attack may be a Botnet formed by a large number of IoT devices making massive DNS requests through a large number of IP addresses, resulting in service interruption [20]. In addition, some attacks are carried out through disguised IP addresses, and these methods cannot effectively defend against DDoS attacks. It is easy to pass attack traffic or block normal traffic due to a single judgment. Therefore, how to reduce the error rate in DDoS attacks is very important.

To sum up, this research is mainly aimed at reducing the misjudgment probability of regular traffic packets and attack traffic packets in DDoS attacks. The convolutional neural network (CNN) is used to distinguish the captured packets to judge whether the current system is normal and use CNN to analyze the difference between a DDoS attack and normal transmission.

The remainder of the present work is structured as follows: Section 2 briefly describes the scenario of a DDoS attack in the IoT network. Section 3 introduces the experimental hardware and architecture, model training dataset, model training methods, IoT architecture, DDoS attack architecture, and system detection architecture. Next, Section 4 is the implementation of the proposed method and experimental outcomes. Finally, Section 5 summarizes the paper.

## 2. Scenarios of DDoS Attack

DDoS is an evolution of Denial of Service attack (DoS), which is a one-to-one transmission mode. The purpose is to send a large number of forged or meaningless packets to the target computer, exhaust the victim's network bandwidth and system resources, stop or interrupt the system services, and prevent other normal users from accessing required resources. However, with the development of computer hardware and network communication, DoS attacks have become more difficult, so the DDoS attack has been developed. It is a botnet composed of two or more hacked computers distributed all over the world, launching a DoS attack on the same target to achieve the purpose of interrupting or stopping the server network service [21], as shown in Figure 1.

A botnet means that hackers use Trojan viruses or system vulnerabilities to write DDoS attack programs, turning other computers into zombie computers (BOT) and forming control nodes that can be used to send forged or spam packets to block the target's network.

### 2.1. Network Bandwidth Consumption DDoS Attacks

Botnets transmit large traffic packets to consume network bandwidth so that the victim's computer is frequently blocked.

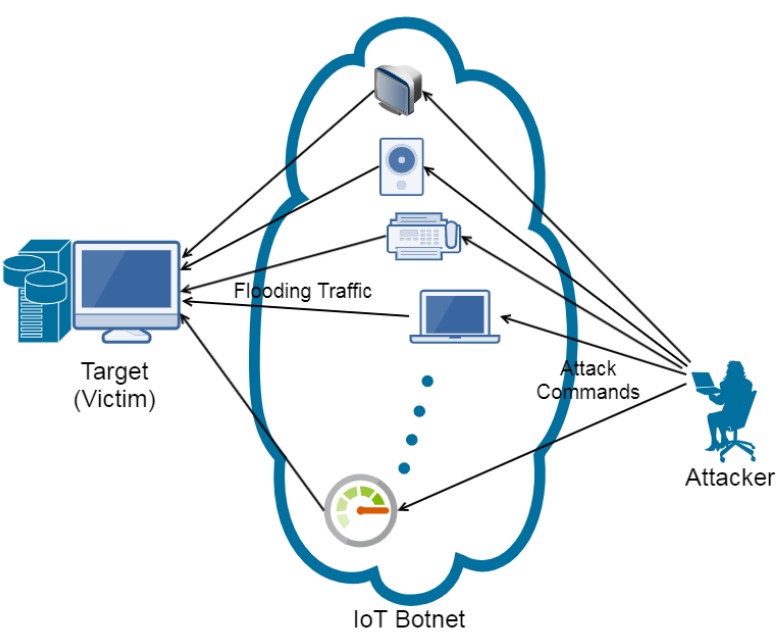

**Figure 1.** Schematic diagram of DDoS attack.

### 2.1.1. User Datagram Protocol (UDP) Flood Attack

UDP is a connectionless transport protocol. When using the UDP protocol to transmit packets, authentication is not required, and a large number of packets can be sent to the victim's computer, which can saturate the bandwidth and make normal services inaccessible. This attack method is shown in Figure 2.

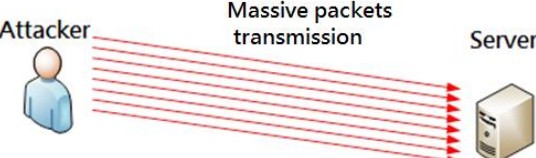

**Figure 2.** UDP flood attack diagram.

### 2.1.2. Internet Control Message Protocol (ICMP) Flood Attack

If the ping command is normally used, the client sends an ICMP echo request header packet to the master, and the master will send an ICMP echo reply header packet to the client to check whether the connection between them can be transmitted properly, as shown in Figure 3. However, the ICMP flood will send a large number of Ping commands to the attacked server in a short time, consuming the host server resources and causing service breakdown, as shown in Figure 4.

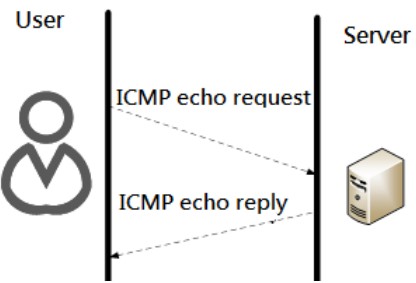

**Figure 3.** Normal ICMP diagram.

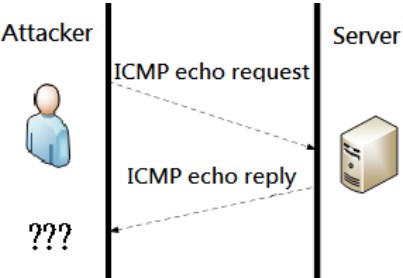

**Figure 4.** ICMP flood attack diagram.

### 2.1.3. Teardrop Attack

Each packet is segmented and shifted before transmission, and the processing information is recorded for future packet reassembly. The teardrop attack will use this method to forge shift information so that the packet cannot be properly reassembled, causing errors.

### 2.2. System Resource Consumption DDoS Attacks

System resource consumption is caused by system transmission vulnerabilities or fake IPs, which exhaust system memory or CPU resources, and eventually lead to service suspension or interruption.

### 2.2.1. Synchronize (SYN) Flood Attack

SYN flood attacks take advantage of the vulnerability of the three-way handshake between the sender and the receiver in the Transmission Control Protocol (TCP). There are two types of SYN flood attacks. The attacker can intentionally not return ACK information or use the fake source IP address in the SYN flood to make the server send SYN + ACK packets to a fake IP address. Because it is a fake IP address, the server cannot receive the ACK packet I response, so the server will keep sending SYN + ACK packets until it times out. This, in turn, consumes server bandwidth and memory resources. The following figure shows the three-way handshake process during normal TCP transmission, as shown in Figure 5. Figure 6 shows the three-way handshake process of TCP transmission during SYN flood attacks.

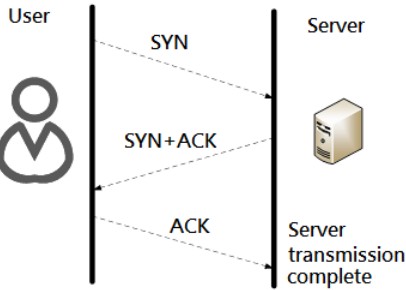

**Figure 5.** TCP normal three-way handshake process.

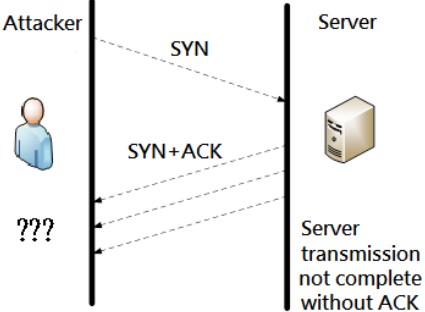

**Figure 6.** SYN flood attack diagram.

### 2.2.2. Local Area Network Denial (LAND) Attack

The main difference with SYN flood is that the fake IP is changed to the same as the attacked host IP. As a result, the host continuously sends SYN + ACK packets back to itself, forming an infinite loop and consuming the resources of the attacked host.

### 2.2.3. DNS Flood Attack

DNS flood is a network attack against the DNS. By sending randomly generated DNS requests to the DNS server through botnets, the server fails to find the relevant subdomain names, thus causing DNS service interruption.

## 3. System Model

This section introduces the experimental hardware and architecture, ML model training dataset, ML model training methods, IoT architecture, DDoS attack architecture, and system detection architecture used in this paper.

### 3.1. Experimental Hardware and Environment Architecture

In this paper, we use one Raspberry pi 3B (802.11n) as the main server for IoT transmission and a wireless network card (2.4 GHz 802.11n/5 GHz 802.11ac) for data transmission with the PC at the edge of computing. The DDoS attack side uses a Raspberry pi 3B and a Raspberry pi 3B + (802.11b/g/n/ac). The transmission between the two is carried out through the Wi-Fi frequency of 2.4 GHz, and system attacks on the data collection end will be carried out through 2.4 GHz Wi-Fi. The mini D1, combined with the temperature and humidity sensor, is used as an IoT sensor node to collect environmental sensor data and transmit data through Wi-Fi at 2.4 GHz. A personal computer is used as the local end of edge computing to monitor whether there is any abnormal transmission in the data collection server. In this paper, the main reason for using edge operation is to reduce the judgment error of the Raspberry pi system and to reduce the transmission delay caused by DDoS attacks on edge operation. Therefore, the practical transmission mode of edge operation is through Wi-Fi in different network domains from the data sensing node and using 5 GHz for transmission. This paper will use two Wireless Access points (AP), AP1 and AP2, as illustrated below. The frequency used by AP1 is 2.4 GHz, and the Wi-Fi frequency used by AP2 is 5 GHz. The Wi-Fi specifications list in Table 1.

**Table 1.** IEEE 802.11 specifications.

| Standard | Frequency (GHz) | Bandwidth (MHz) | TX Rate Mbit/s | MIMO |
|---|---|---|---|---|
| IEEE 802.11 | 2.4 | 20 | 2 | NA |
| IEEE 802.11a | 5 | 20 | 54 | NA |
| IEEE 802.11b | 2.4 | 20 | 11 | NA |
| IEEE 802.11g | 2.4 | 20 | 54 | NA |
| IEEE 802.11n | 2.4/5 | 40 | 150 | 4 |
| IEEE 802.11ac | 2.4 | 160 | 866.7 (single stream) | 8 |
| IEEE 802.11ax | 2.4/5/6 | 160 | 1120 (single stream) | 8 |

### 3.2. Training Dataset and Model Training

In this paper, an additional Wi-Fi wireless network card is installed on the Raspberry pi that collects the values of the sensing nodes so that it can have two IPs at the same time, one is the wireless transmission IP of the Raspberry pi called IP1, and the IP provided by the wireless card is called IP2. IP1 and IP2 will be used as per the description below. IP1 is used to receive information about the sensing node with a Wi-Fi frequency of 2.4 GHz, while IP2 is used only for the connection to the edge computing computer. In this paper, the number of packets per second and the characteristics of each packet flow in the DDoS attack on the server is collected as the experimental dataset for training models of CNNs and neural-like networks.

3.2.1. Packet Traffic Capture Dataset

This dataset is based on the results of the paper in [22]. The CPU and memory usage rate of the system hardware will be affected while suffering from attacks. The total number of packet traffic per second will be counted, and the usage rate of the CPU and memory will be measured under monitoring by adding TCP, UDP, ICMP, and other transmission packets. In addition, normal transmission, SYN flood attack, UDP flood attack, ICMP flood attack, and MIX flood attack are recorded, respectively. During each test, the packet traffic capture time is 2 h per test, and the system is restarted to ensure that the data is not affected by the previous attack tests. The packet traffic features capturing and delivering diagram is shown in Figure 7. The packet traffic features captured directly from the packet information are usage rates of CPU (CPU) and memory (Memory), the numbers of TCP packet (No. TCP), UDP packet (No. UDP), ICMP packet (No. ICMP), and other packets (No. Other). The packet traffic dataset under normal transmission is shown in Table 2.

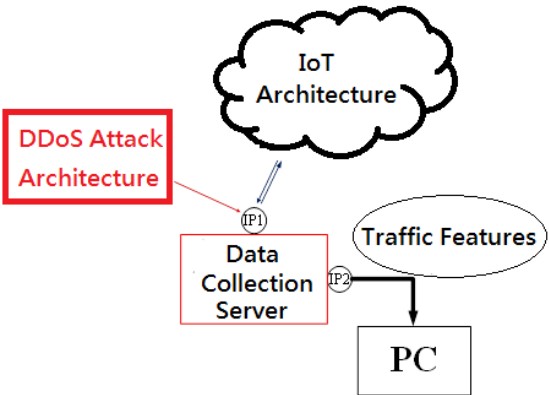

**Figure 7.** Schematic diagram of packet traffic features capturing and packet delivering.

**Table 2.** The packet traffic dataset.

| Memory | CPU | No. TCP | No. UDP | No. ICMP | No. Other |
|---|---|---|---|---|---|
| 33.6351795 | 6.1 | 0 | 1 | 0 | 0 |
| 33.6638273 | 13.2 | 0 | 2 | 0 | 0 |
| 33.6333890 | 0 | 0 | 1 | 0 | 0 |
| 33.6329414 | 6.7 | 10 | 2 | 0 | 0 |
| 33.6387604 | 0 | 10 | 1 | 0 | 0 |
| 33.6324937 | 0 | 10 | 2 | 0 | 0 |
| 33.6602463 | 5.8 | 10 | 1 | 0 | 0 |
| 33.6602463 | 0.8 | 10 | 2 | 0 | 0 |
| 33.66.2463 | 6.9 | 10 | 1 | 0 | 0 |
| 33.6338366 | 10.8 | 10 | 1 | 0 | 0 |

3.2.2. Packet Features Capture Dataset

This dataset is based on the results of the paper in [22]. It captures the information of each packet flowing through IP1 by TShark and transmits the captured packet information to the edge computing computer through IP2, as shown in Figure 8. The computer will distinguish the packet transmission mode, the interval time between two packet transmissions, the sequence number, and the captured transmission packet size from the captured information. The pre-processing process performed by the edge computing computer to capture the packet features is shown in Figure 9. The interval between two packets (interval), sequence number (sequence), and packet size (size) will be captured directly from the packet information. If the field is empty in packet, information under specific conditions will be filled 0, and the transmission mode (TX mode) will be coded in 0, 1, and 2 for TCP, UDP, and ICMP, respectively. The preprocessing result from a packet information

is shown in Figure 10. The packet features dataset under normal transmission is shown in Table 3.

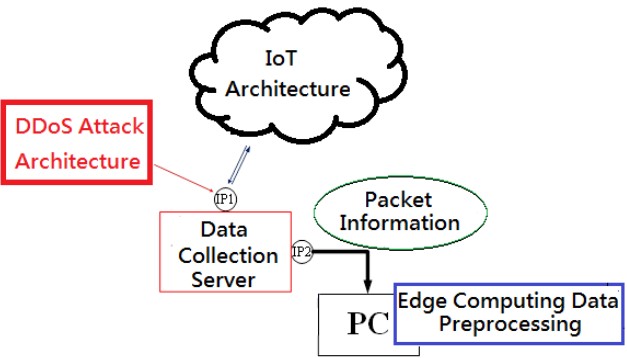

**Figure 8.** Schematic diagram of packet features capturing and packet delivering for edge computing.

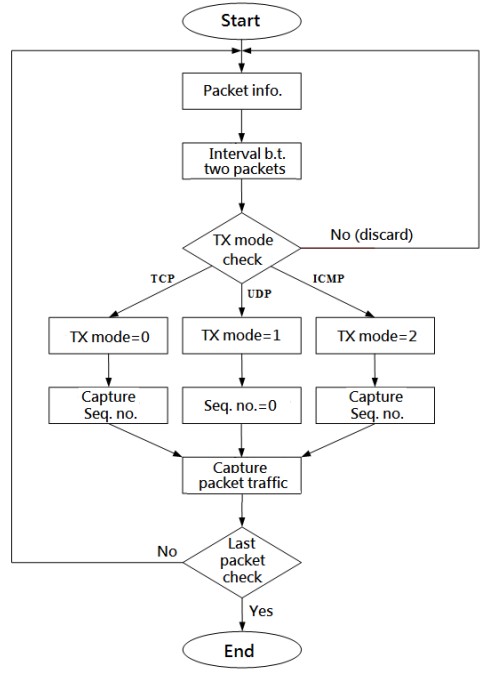

**Figure 9.** The procedure of data preprocessing for packet features capturing.

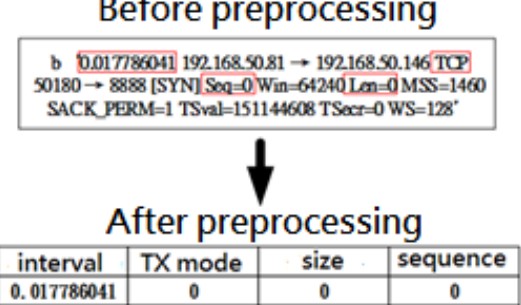

**Figure 10.** The result of data preprocessing for packet feature capture.

**Table 3.** The packet features dataset.

| Interval | TX Mode | Size | Sequence |
|---|---|---|---|
| 1.011111562 | 1 | 4 | 0 |
| 0.000112396 | 2 | 4 | 0 |
| 0.500199167 | 0 | 16 | 1 |
| 0.001090100 | 0 | 0 | 1 |
| 0.000265677 | 0 | 6 | 1 |
| 0.004458860 | 0 | 0 | 7 |
| 0.009158385 | 0 | 0 | 17 |
| 0.000061667 | 0 | 0 | 8 |
| 0.000308073 | 0 | 0 | 0 |
| 0.000065312 | 0 | 0 | 0 |

*3.3. IoT Architecture*

The sensor node used in this paper is a temperature and humidity sensor (DHT11) for sensing the ambient temperature and humidity, and its sampling range is 0–50 °C with a measurement error of ±5 °C, 20–90% RH with a measurement error of ±5% RH and a sampling time of 1 time/s. The data is transmitted through D1 mini, IP1, and AP1 for Wi-Fi connection with a frequency of 2.4, as shown in Figure 11, and the data is stored in the server. The transmission process is shown in Figure 12.

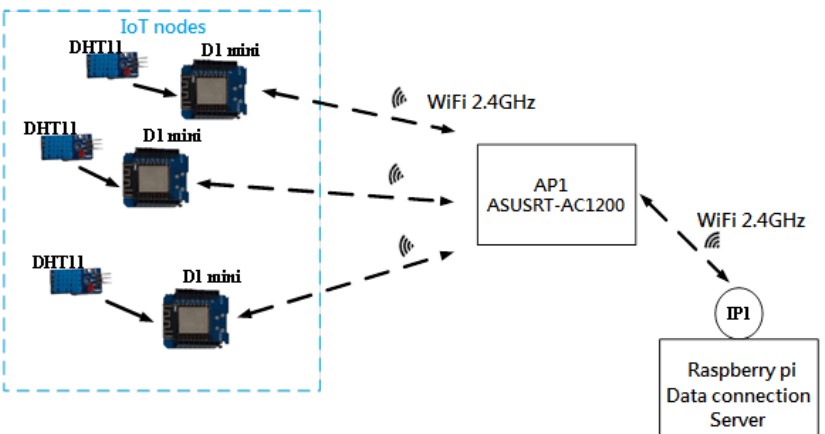

**Figure 11.** The architecture of IoT in the real experiment system.

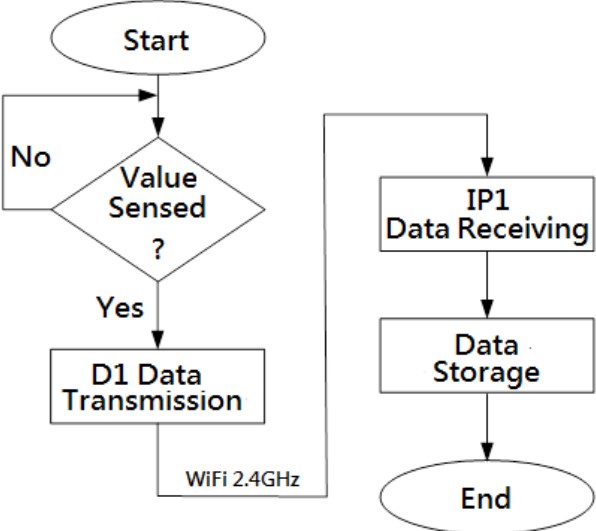

**Figure 12.** Diagram of the transmission process.

### 3.4. DDoS Attack Architecture

To collect the status of the IoT server under a DDoS attack, this paper sets up a DDoS attack environment. The DDoS tool used in this paper is TFN2K, which is used to simulate the attack on the server. Its architecture is shown in Figures 13 and 14, and this tool can launch SYN flood attacks, UDP flood attacks, ICMP flood attacks, and mixed attacks (MIX flood) on the server, respectively. This tool can operate other hosts through control commands to form a botnet to launch DDoS attacks. Therefore, this paper uses the four attack methods available to the tool and normal transmission for data collection. There are 25,000 messages for packet traffic data capture and 4.5 million messages for packet feature data capture.

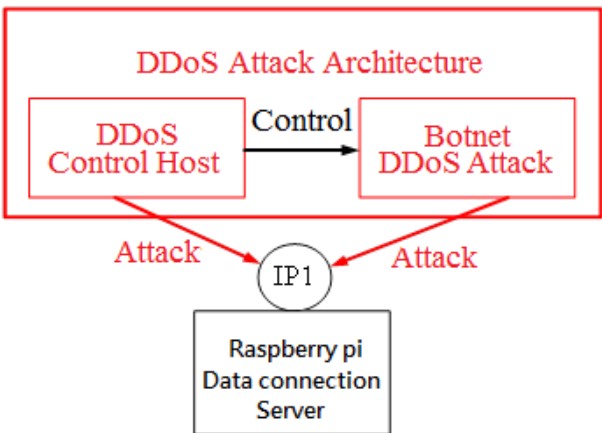

**Figure 13.** Diagram of DDoS attack for the practical evaluation system.

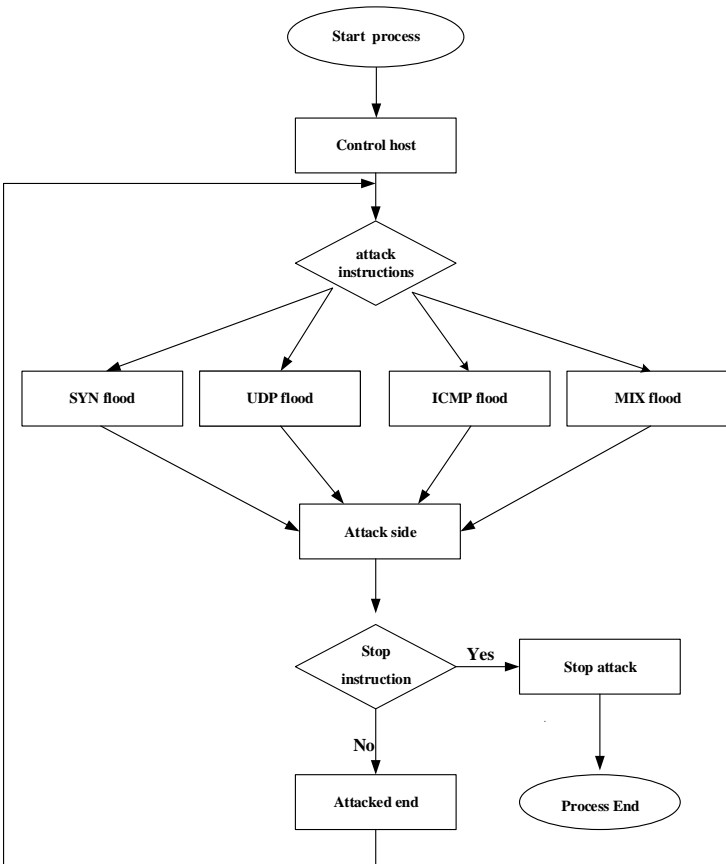

**Figure 14.** The procedure of DDoS attack.

### 3.5. System Detection Architecture

The system detection in this paper is to capture the packet traffic passing through IP1 and analyze the packet features, and then transmit it to the edge computing computer through IP2 to determine whether the system is abnormal. IP1 is connected to AP1 for the transmission of the IoT system, so it is at a high risk of attack. Therefore, this study will attack IP1 to simulate the server being attacked. Due to the defects in the Raspberry pi hardware system that may lead to misjudgment, a wireless network card is added to the data server in this paper, which is connected to AP2 to communicate with the edge computer. Therefore, the IP of the additional network card is called IP2, and IP2 is only used for Communicating with edge computers. The detection process is shown in Figure 15.

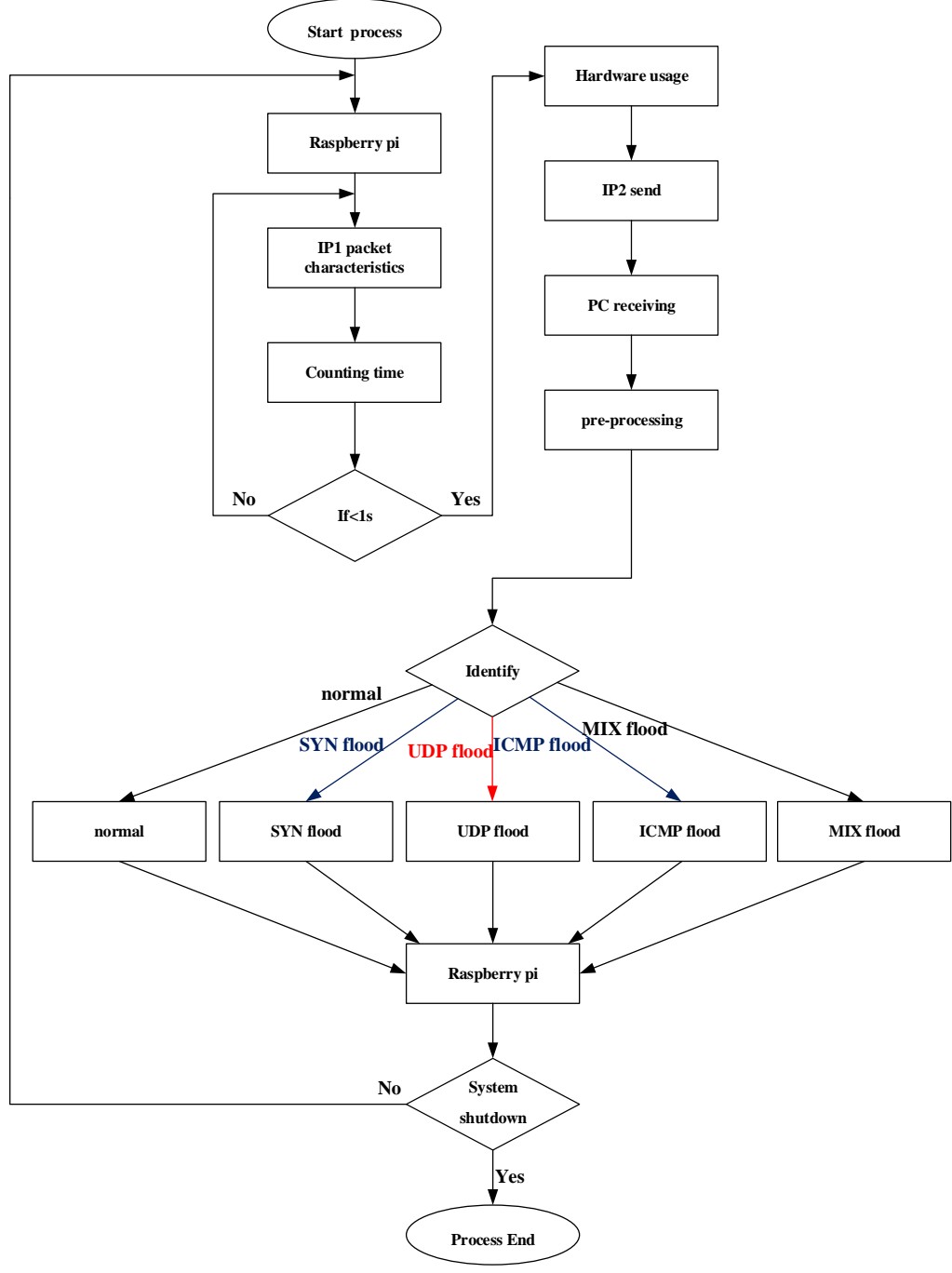

**Figure 15.** Procedure from packet receiving to edge computing.

## 4. Results and Discussion

In this paper, TFN2K is used to collect packet traffic statistics, CPU and memory usage, average network speed, and other characteristics for five scenarios, including normal transmission, SYN flood attack, UDP flood attack, ICMP flood attack, and MIX flood attack. The collected data are preprocessed to build a packet traffic dataset and packet feature dataset, respectively, which can be used as the data for artificial intelligence model training. In this paper, the packet traffic is monitored for five minutes, and the relevant features of the packet traffic during this period are captured. Figure 16a–e shows the total number of TCP, UDP, ICMP, and other packets under the normal transmission, SYN flood attack, UDP flood attack, ICMP flood attack, and MIX flood attack, respectively. The ordinate is the cumulative total number of packets, and the abscissa is the number of counts per 3 s.

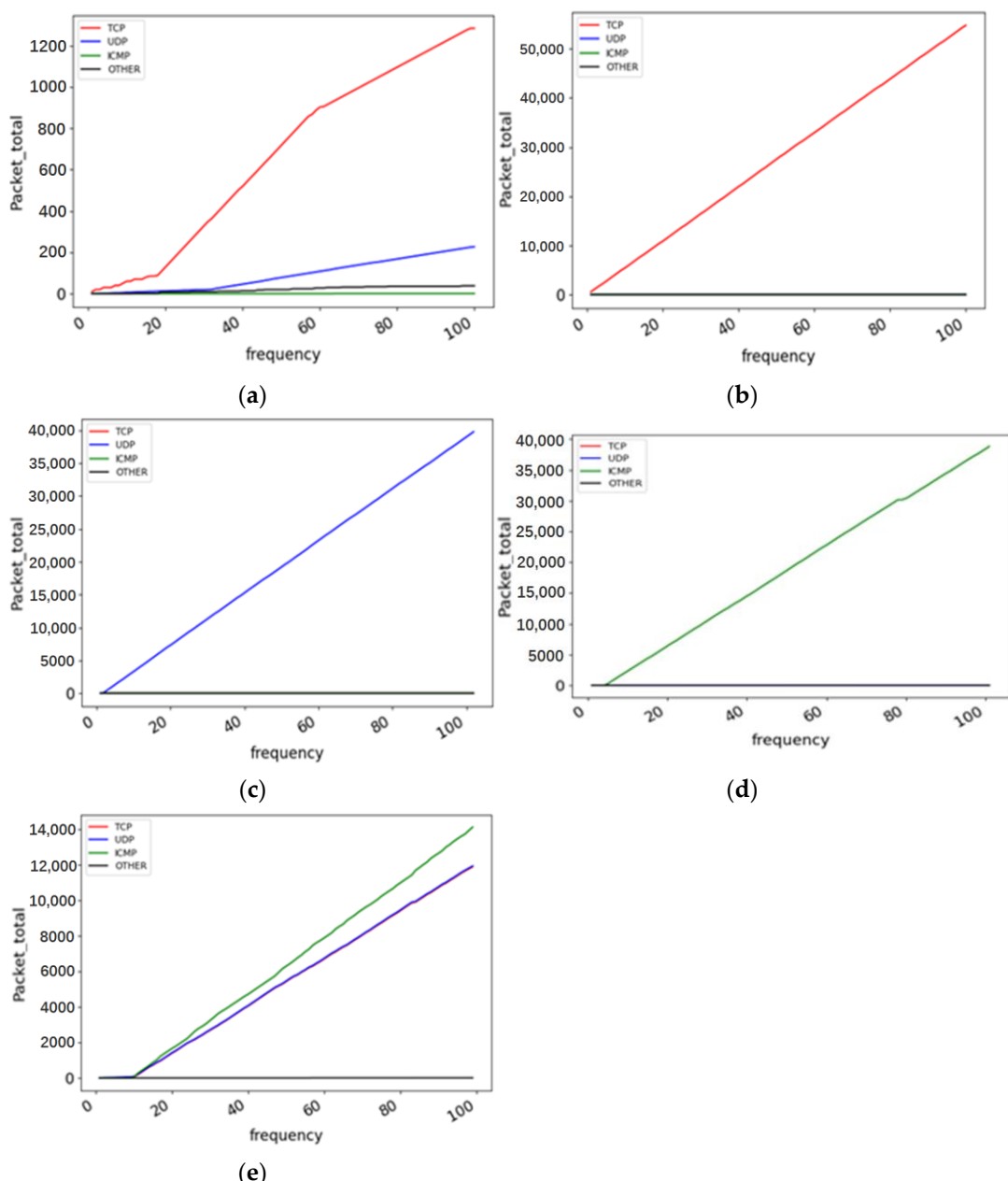

**Figure 16.** Packet flow cumulative diagram. (**a**) Normal transmission; (**b**) SYN flood attack; (**c**) UDP flood attack; (**d**) ICMP flood attack; (**e**) MIX flood attack.

Figure 16a–e shows the total number of TCP, UDP, ICMP, and other packets under the normal transmission, SYN flood attack, UDP flood attack, ICMP flood attack, and MIX flood attack, respectively. From Figure 16a, it can be seen that under normal transmission conditions, there are all kinds of packets, and the cumulative number of packets increases moderately. However, no matter what kind of flood attack, the cumulative number of packets increases suddenly. Moreover, there are different cumulative numbers for different packet types when under attack. These characters can not only be used to judge whether an attack has been suffered but also can be used to identify the type of attack.

The experimental retrieval time is 5 min, so there are 100 records. Compared with Figure 16a, it can be seen in Figure 16b that suffered from an SYN flood attack, its TCP packets increased a lot, and the total number of TCP packets flowing through IP1 within 5 min was about 50,000, and other types of packets are almost equal to 0. In Figure 16c of the UDP flood attack, it can be seen that UDP packets have increased a lot, while other types of packets are almost equal to 0. The collection time is 5 min. It can be seen that the sum of UDP packets during this period reached about 40000. In Figure 16d of the ICMP flood attack, it can be seen that ICMP packets increased a lot, while other types of packets are almost equal to 0, and the collection time is 5 min. During this period of time, the sum of ICMP packets also reached about 40,000. Figure 16e shows the situation of being attacked by other packets (MIX flood attack). At this time, the server flows through a large number of various types of packets, and the number of packets with UDP and TCP characteristics is almost the same. Therefore, in our experiment, it is obvious to observe that when attacked, the increase of various packets with this time and the change of the number of packets.

Figure 17a–e are the usage rates of their CPU and memory, respectively. As can be seen from Figure 17a, the CPU usage rate changes little during normal transmission, but it will fluctuate greatly under attacks. Similarly, the memory usage rate is much lower under normal transmission than under attacks. It helps to detect whether the system is under attack. In Figure 17b, it can be seen that when an SYN flood attack is encountered, the usage rate of CPU is about 20% higher than that in Figure 17a under normal conditions. As shown in Figure 17c, when suffering from UDP flood attacks, the CPU usage rate fluctuates violently, and its highest usage rate reaches more than 50%. In Figure 17d, we can see that when the ICMP flood attack is encountered, the CPU's usage rate fluctuates violently. The situation of the oscillation is the same as that of the UDP flood attack, but it has different characteristics. In Figure 17e, it can be seen that when the MIX flood is encountered During the attack, the CPU's usage rate will suddenly rise and fluctuate violently, just like the previous attacks. As shown in Figure 17b–e, the usage rate of the memory remains roughly the same as normal. Therefore, we can find out its characteristics from the fluctuation of CPU usage. From the results in Table 4, it can be seen that various attacks will affect the network speed during transmission. In our experiment, the average network transmission speed will drop from an average of 14.2 Mbps to about 0.3 Mbps. Based on the above experimental results, we use a neural network for edge computing and model training.

**Table 4.** Average speed under the normal transmission, SYN flood attack, UDP flood attack, ICMP flood attack, and MIX flood attack.

| Condition | Value |
|---|---|
| Normal Transmission | 14.2 Mbps |
| SYN Flood Attack | 0.30 Mbps |
| UDP Flood Attack | 0.30 Mbps |
| ICMP Flood Attack | 0.31 Mbps |
| MIX Flood Attack | 0.31 Mbps |

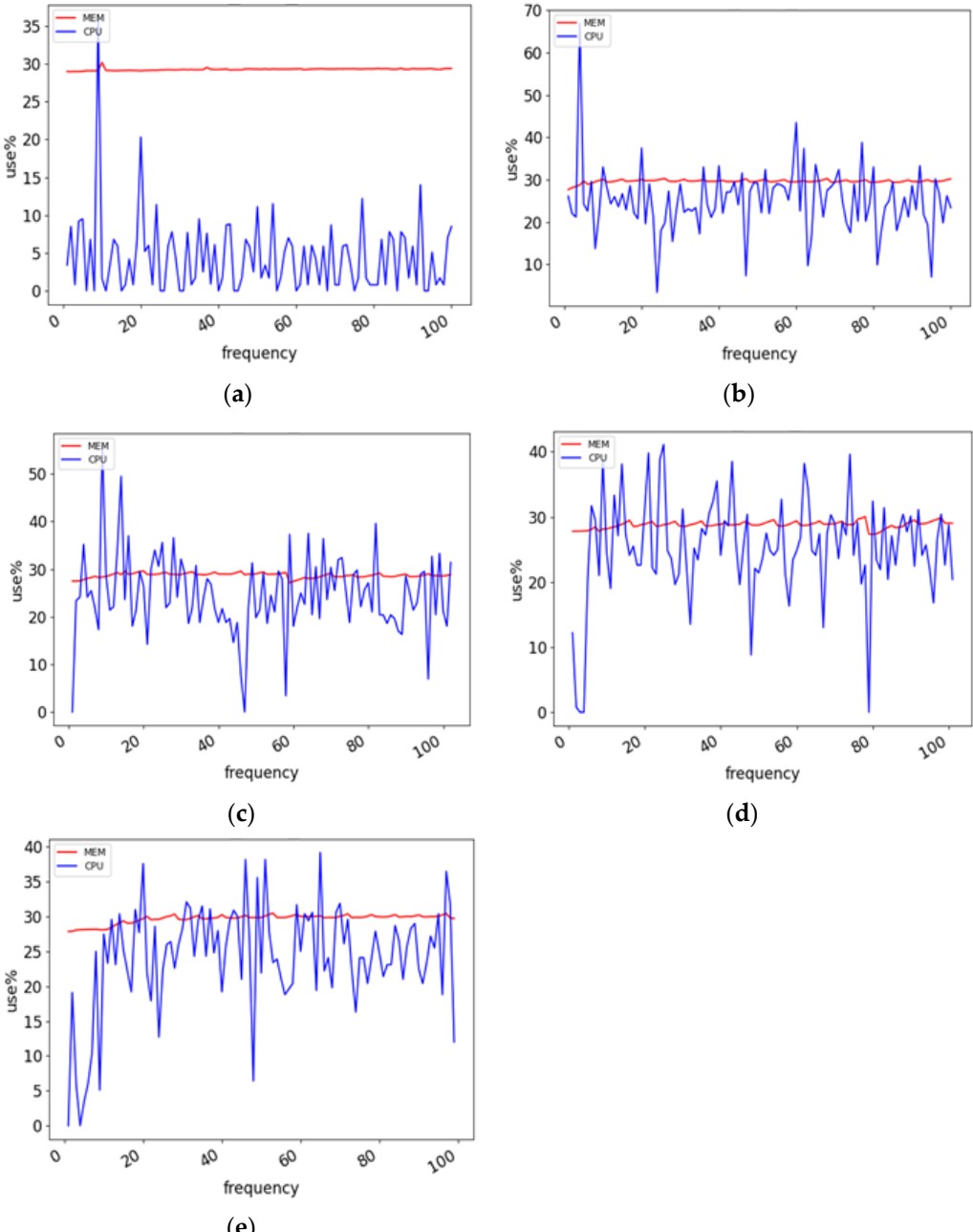

**Figure 17.** CPU and memory usages (**a**) normal transmission; (**b**) SYN flood attack; (**c**) UDP flood attack; (**d**) ICMP flood attack; (**e**) MIX flood attack.

In this paper, the model of a NN, one-dimensional CNN, and two-dimensional CNN are respectively trained and evaluated. The best-performing model will be applied to the practical validation of the proposed system. Accuracy is used to evaluate the proportion of correct predictions of true and false under all conditions, which can be expressed as

$$\text{Accurary} = \frac{\text{TP} + \text{TN}}{\text{TP} + \text{TN} + \text{FP} + \text{FN}} \times 100\% \tag{1}$$

where the four conditions, True Positive (TP), True Negative (TN), False Positive (FP), and False Negative (FN), are described in Table 5.

**Table 5.** Definition of TP, FP, FN, and TN.

| Transmission Mode | AI Model Identification Results | |
| --- | --- | --- |
| | Normal Transmission | Abnormal Transmission |
| Normal | TP | FN |
| Abnormal | FP | TN |

In this paper, we use the packet traffic dataset to train the three models mentioned above. One data is used as the labeled input, and the training parameters are 10, 50, 100, 150, and 200 times respectively. The batch size of each training is 2500, and the learning rate is 0.00001. The model uses three hidden layers, 128, 64, and 32 neurons, respectively, with an excitation function of "ReLU" and the output layer using the "softmax" as an excitation function for classification. When using the packet feature dataset for training, four pieces of data are used as label input and trained 10, 20, 30, 40, and 50 times respectively. The batch size is 2500, and the learning rate is 0.00001. The model has three hidden layers of 128, 64, and 32 neurons, respectively, and the excitation function is "ReLU." The excitation function "softmax" is used for classification in the output layer. For one-dimensional and two-dimensional CNN, a dropout layer is added after the first layer to prevent overfitting.

The training accuracies of the two different pieces of training are shown in Tables 6 and 7, respectively. It can be seen from Tables 6 and 7 that with the increase in the number of training times and the ratio of input, the accuracy of training can reach more than 99% in the end. This shows that according to the characteristics of the environment, there must be more than a certain amount of data collected in order to have good training results. As shown in Tables 6 and 7, the two-dimensional CNN model is more accurate after training, both in terms of packet traffic and feature training, named traffic model and feature model, respectively hereafter, so it will be used as the identification model for the practical validation.

**Table 6.** Accuracy for NN, 1D CNN, and 2D CNN trained by using packet traffic dataset.

| Model | Number of Training | | | | |
| --- | --- | --- | --- | --- | --- |
| | 10 | 50 | 100 | 150 | 200 |
| NN | 49.9% | 83.7% | 99.2% | 99.1% | 99.0% |
| 1D CNN | 50.9% | 53.4% | 98.8% | 99.1% | 99.1% |
| 2D CNN | 15.4% | 43.0% | 99.3% | 99.2% | 99.5% |

**Table 7.** Accuracy for NN, 1D CNN, and 2D CNN trained by using packet feature dataset.

| Model | Number of Training | | | | |
| --- | --- | --- | --- | --- | --- |
| | 10 | 20 | 30 | 40 | 50 |
| NN | 58.0% | 86.0% | 98.9% | 99.3% | 99.7% |
| 1D CNN | 93.5% | 95.5% | 99.7% | 99.8% | 99.8% |
| 2D CNN | 97.4% | 99.5% | 99.8% | 99.8% | 99.8% |

Both the correct rate and loss rate of the two-dimensional CNN for the traffic model and feature model is shown in Figures 18 and 19, respectively.

In this paper, we use a trained two-dimensional CNN model for real experiment verification. The data collection server IP1 is subjected to normal transmission, SYN flood attack, UDP flood attack, ICMP flood attack, and MIX flood attack, respectively, and the number and feature of packets flowing through IP1 are collected. It will be sent to the edge computing computer through IP2 to identify the current situation of the server by using a trained model. The edge computing computer will preprocess the received data and input them into the traffic model and feature model, respectively, for identification. The final identification result is obtained by combining the obtained identification rates and

their weights. Considering that the data of the packet traffic dataset may be affected by the increasing number of users in the IoT, and the features captured will not be able to identify because the data captured is too small, this paper gives the traffic model and feature model the weights of 70% and 30% respectively, as the basis for judging the identification results.

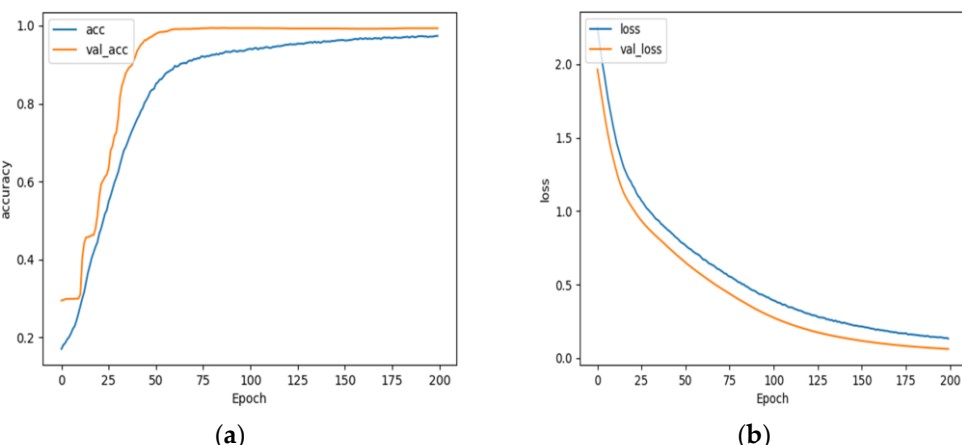

**Figure 18.** (**a**) Accuracy rate; (**b**) Loss rate of the two-dimensional CNN for the packet traffic model.

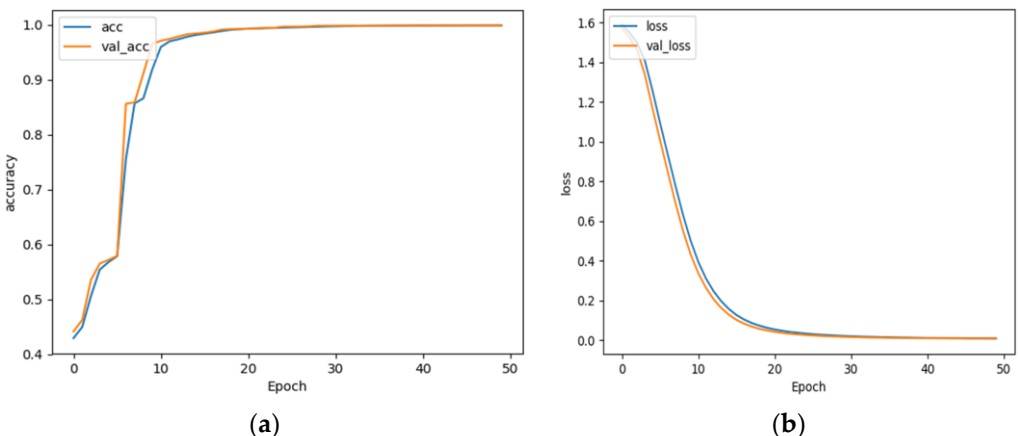

**Figure 19.** (**a**) Accuracy rate, (**b**) loss rate of the two-dimensional CNN for the packet feature model.

Table 8 lists the identification results on the edge computer when measured under different conditions, in which Feature is the identification result of the feature model, Flow means the identification result of the traffic model, and Weighted is the identification result after individual weighting on the two models. According to the table, the proposed architecture of edge computing with a trained CNN model can make correct identifications under the normal transmission, SYN flood attack, UDP flood attack, ICMP flood attack, and MIX flood attack, respectively. Even in the lack of packets case during normal transmission, the feature model cannot recognize it normally, but the traffic model can still recognize it when the data is rare. Since the traffic model accounts for 70% of the weight, the weighted identification result is normal. The above experimental results are mainly because this system adopts two independent two-dimensional CNN models trained by the packet traffic dataset and packet features dataset, respectively, and recognizes the existing situation of the system at the same time and then adds appropriate weights to them as a basis for judgment. This design can maximize the use of information about changes in related characteristics when packets are transmitted in the system, which can effectively improve the accuracy of recognition and make up for the shortcomings of using a single model for judgment.

**Table 8.** Time interval identification under the normal transmission, SYN flood Attack, UDP flood Attack, ICMP flood Attack, and MIX flood Attack.

| Condition | Feature | Flow | Weighted |
|---|---|---|---|
| Normal Transmission (lack of packets) | Lack data | Normal | Normal |
| Normal Transmission | Normal | Normal | Normal |
| SYN Flood Attack | SYN Flood | SYN Flood | SYN Flood |
| UDP Flood Attack | UDP Flood | UDP Flood | UDP Flood |
| ICMP Flood Attack | ICMP Flood | ICMP Flood | ICMP Flood |
| MIX Flood Attack | MIX Flood | MIX Flood | MIX Flood |

Table 9 shows the time interval, named identification time, between Raspberry pi transmitting the captured data to the edge computing computer and receiving the identification result returned by the edge computing computer. From this table, it can be concluded that the identification time does not cause the system identification delay due to the DDoS attack.

**Table 9.** The identification time under the normal transmission, SYN flood Attack, UDP flood Attack, ICMP flood Attack, and MIX flood Attack.

| Condition | Identification Time (s) |
|---|---|
| Normal Transmission | 8.12 |
| SYN Flood Attack | 8.00 |
| UDP Flood Attack | 8.72 |
| ICMP Flood Attack | 8.12 |
| MIX Flood Attack | 8.17 |

## 5. Conclusions

This paper proposes a DDoS attack detection system based on edge computing. The edge computing computer in this system uses a trained two-dimensional CNN model to identify whether the data collection server in the IoT is under a DDoS attack and how the attack is conducted, and immediately notify the data collection server to reduce the impact of DDoS attacks on the data transmission in the IoT system.

The proposed DDoS detection system can be built properly without changing the original IoT hardware structure due to adopting the edge computing architecture. In addition, two two-dimensional CNN models trained by packet traffic and packet features data, respectively, are used simultaneously to identify DDoS attacks that can effectively improve the accuracy of identification. However, the two-dimensional CNN models only trained for usual DDoS attacks will cause reducing the accuracy of identification for new types of DDoS attacks. Moreover, the datasets used in the experiments in this paper were completely captured from the experimental network, which will result in the result of the paper falling into a special case. Thus, using an open dataset to verify this system is necessary for the future.

**Author Contributions:** Conceptualization, S.-H.L., C.-H.C. and Y.-F.H.; Investigation, Y.-H.L., Y.-L.S., and C.-H.C.; Methodology, S.-H.L., C.-H.C., Y.-H.L. and Y.-F.H.; Software, Y.-L.S., Y.-H.L. and C.-H.C.; Validation, C.-H.C.; Supervision, C.-H.C. and Y.-F.H.; Writing—original draft, Y.-L.S., S.-H.L. and C.-H.C.; Writing—review and editing, S.-H.L., C.-H.C. and Y.-F.H. All authors have read and agreed to the published version of the manuscript.

**Funding:** This research was funded by the Ministry of Science and Technology (MOST), R.O.C. grant number MOST 111-2221-E-324-018 and MOST-111-2637-E-150-001.

**Institutional Review Board Statement:** Not applicable.

**Informed Consent Statement:** Not applicable.

**Data Availability Statement:** Not applicable.

**Conflicts of Interest:** The authors declare no conflict of interest.

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
