# Peer review of "Detection and Prevention of DDoS Attacks on the IoT"

_applsci, doi:10.3390/app122312407_

Round 1

Reviewer 1 Report

The paper suggested deep learning-based techniques for detecting and preventing numerous forms of distributed denial-of-service (DDoS) attacks on two generated datasets using internet of things (IoT) devices integrated with edge computing. By employing Raspberry pi, DHT11 sensor, and edge computing approach, packet traffic and packet feature datasets were created via the use of two IP addresses by the constructed IoT system. After appropriate pre-processing, a neural network, 1D, and 2D CNN models were used to detect SYN, UDP, ICMP, and MIX flood DDoS attacks. The result from the 2D CNN model proved to be better. 

 Comments

1)      The abstract section needs to present the main findings of the research work precisely.

2)      Sufficient recent and relevant reviewed pieces of literature related to the topic should be provided. A separate section should be created to accommodate the reviewed literature if necessary.

3)      Several details regarding the two datasets still need to be provided. Details like the pre-processing technique employed and the percentage of data used for training and validation to ensure the generalizability of the proposed models should be supplied.

4)      The authors(s) should ensure to state the challenges encountered and the limitations of the suggested system to give room for future research.

5)      Ensure to provide adequate references and citations based on the IEEE standards to make the paper more scholarly.

6)      The manuscript requires English language grammar editing. Several sentences are hard to comprehend fully. The Author(s) is advised to reach out to editors or use editing tools for corrections regarding the use of language.

7)      Abbreviations used in the study should be given in their long-form only where they are first used. There is no need to use the long forms again later. For example, the Internet of Things and denial of service, convolutional neural network, etc.

8)      The advantages and disadvantages of the study should be explained in detail in order to guide the reader.

As such, the study presents a small simulation for the detection of distributed denial of service attacks. The study does not contribute enough to the scientific literature. For example, it is not clear how much bandwidth the work will be sufficient for the attack that can be made. Again, it is not clear from the model point of view how many nodes or how many simultaneous attacks the study can detect. It is not clear that the work as such would protect a real system or how much bandwidth it would protect. The original value of the study is unfortunately weak due to the deficiencies it contains. I do not think this version of the work is acceptable.

Author Response

Response to Reviewer 1 Comments

The paper suggested deep learning-based techniques for detecting and preventing numerous forms of distributed denial-of-service (DDoS) attacks on two generated datasets using internet of things (IoT) devices integrated with edge computing. By employing Raspberry pi, DHT11 sensor, and edge computing approach, packet traffic and packet feature datasets were created via the use of two IP addresses by the constructed IoT system. After appropriate pre-processing, a neural network, 1D, and 2D CNN models were used to detect SYN, UDP, ICMP, and MIX flood DDoS attacks. The result from the 2D CNN model proved to be better. 

Point 1: The abstract section needs to present the main findings of the research work precisely.

Response 1: Thank you very much for this comment. The abstract has been modified to present the main findings of the research.

“In this paper, we proposed an autonomous defense system that combines edge computing with a 2D convolutional neural network (CNN) to recognize whether the data server in IoT suffers from DDoS attacks and identify the attack mode. The accuracy of trained 2D CNN is up to 99.5% and 99.8% for packet traffic and packet features training respectively. A field experiment results show that the data server in the proposed system can effectively distinguish the difference between the DDoS attacks and the normal transmission to reduce the impact of DDoS attacks on the IoT data storage while it is under attack.”

Point 2: Sufficient recent and relevant reviewed pieces of literature related to the topic should be provided. A separate section should be created to accommodate the reviewed literature if necessary.

Response 2: Thank you very much for this comment. A recent and relevant literature review has been added in Introduction Section.

“Some relevant pieces of literature about machine learning (ML) based methods to mitigate DDoS attacks in IoT systems in recent years will be reviewed below. An evaluation of using a Random Neural Network (RNN) trained by normal traffic data compared to the Long-Short Term Memory (LSTM) to detect SYN flood DDoS attack was provided in [13]. It shows better results in RNN than LSTM, but its accuracy was almost 81% and not considered better enough to rely on. In [14] presented a new deep neural network to identify the network flows being normal or abnormal. The authors adopted a feedforward back-propagation design with seven secret layers and tested the method for DDoS detection using the most up-to-date Canadian data set (CIC IDS 2017). The test provided a value of 0.99 scores which means that the experimental results were accurate in terms of Recall and Precision. A resource-friendly ML algorithm called Edge2Guard (E2G) was introduced in [15]. It was trained and tested over the N-BaIoT dataset of normal and attack network traffic logs recorded by using Mirai and Bashlitte Botnets. The algorithm has re-source-friendly detection depending on creating an E2G model for each MCU-based IoT device separately in the system. The disadvantage of this algorithm is that model should be upgraded frequently after being trained with data from the developed type of malware action resulting in rising difficulties in the deployment process. An effective method employing two essential attributes named Volumetric and Asymmetric to detect two forms of flooding-based DDoS attacks was presented in [16]. The proposed DDoS attack detection method based on SDN can cause minimal disruption to effective user activity and reduce both training and testing time. In addition, it proposed the Advanced Support Vector Machine (ASVM) technique to enhance the current Support Vector Machine (SVM) algorithm to detect DDoS flooding attacks effectively. In [17] proposed a new detection classification system based on SVM and CNN ML algorithms. It converts the binary files into visualized images in grayscale. Then the CNN and SVM process these images to detect if a file contains maliciously injected code. The accuracy of this method is up to 94% in the binary classification case but only 81% in the multi-classification case. A new ML method based on clustering and graph structure features to predict the occurrence of DDoS attacks was provided in [18]. The method creates the edge and vertex structures in graph theory and extracts eight features of traffic data as input variables. Then uses the principal component analysis (PCA) model to extract the features of DDoS and normal communication. Finally, the fuzzy C-means (FCM) clustering method is used to detect DDoS. The availability of this method is verified by taking 2000 traffic data in CICIDS-2017 as an example. The results of recall, false positive, true positive, true negative, and false negative were 100.00%, 1.05%, 68.95%, 0.00%, and 30.00%, respectively, indicating that it improves the reliability of detection and has a good detection effect on DDoS attacks compared to other methods.”

Point 3: Several details regarding the two datasets still need to be provided. Details like the pre-processing technique employed and the percentage of data used for training and validation to ensure the generalizability of the proposed models should be supplied.

Response 3: Thank you very much for this comment. More details about the two datasets have been added in Sections 3.2.1 and 3.2.2.

In Section 3.2.1:

“This dataset is based on the results of the paper in [22]. The CPU and memory usage rate of the system hardware will be affected while suffering from attacks. The total number of packet traffic per second will be counted and the usage rate of the CPU and memory will be measured under monitoring by adding TCP, UDP, ICMP, and other transmission packets. In addition, normal transmission, SYN flood attack, UDP flood attack, ICMP flood attack, and MIX flood attack are recorded respectively. During each test, the packet traffic capture time is 2 hours per test, the system is restarted to ensure that the data is not affected by the previous attack tests. The packet traffic features capturing and delivering diagram is shown in Figure 7. The packet traffic features captured directly from the packet information are usage rates of CPU (CPU) and memory (Memory), the numbers of TCP packet (No. TCP), UDP packet (No. UDP), ICMP packet (No. ICMP) and other packets (No. Other). The packet traffic dataset under normal transmission is shown in Table 2.

Table 2. The packet traffic dataset.

Memory

CPU

No. TCP

No. UDP

No. ICMP

No. Other

33.6351795

6.1

0

1

0

0

33.6638273

13.2

0

2

0

0

33.6333890

0

0

1

0

0

33.6329414

6.7

10

2

0

0

33.6387604

0

10

1

0

0

33.6324937

0

10

2

0

0

33.6602463

5.8

10

1

0

0

33.6602463

0.8

10

2

0

0

33.66.2463

6.9

10

1

0

0

33.6338366

10.8

10

1

0

0

In Section 3.2.2:

“The interval between two packets (interval), sequence number (sequence), and packet size (size) will be captured directly from the packet information, if the field is empty in packet information under specific conditions will be filled 0, and the transmission mode (TX mode) will be coded in 0, 1 and 2 for TCP, UDP, and ICMP respectively. The preprocessing result from a packet information is shown in Figure 10. The packet features dataset under normal transmission is shown in Table 3.

Table 3. The packet features dataset.

Interval

TX mode

Size

Sequence

1.011111562

1

4

0

0.000112396

2

4

0

0.500199167

0

16

1

0.001090100

0

0

1

0.000265677

0

6

1

0.004458860

0

0

7

0.009158385

0

0

17

0.000061667

0

0

8

0.000308073

0

0

0

0.000065312

0

0

0

Point 4: The authors(s) should ensure to state the challenges encountered and the limitations of the suggested system to give room for future research.

Response 4: Thank you very much for this comment. The main limitation of the proposed system is the acquirement of packet traffic data and packet features data for training the 2D CNN model. The performance verification in this paper is made by an experiment network created practically, not just through total software simulations. Therefore, the training data used in the paper may be only feasible for the experiment network. This limitation has been stated as a disadvantage in Section 5.

Point 5: Ensure to provide adequate references and citations based on the IEEE standards to make the paper more scholarly.

Response 5: Thank you very much for this comment. The references in the revised version have referred to more IEEE literature by adding the refs. [15-17].

[15] B. Sudharsan; D. Sundaram; P. Patel; J. G. Breslin; M. I. Ali. Edge2Guard: Botnet attacks detecting offline models for resource-constrained IoT devices. In Proceedings of 2021 IEEE International Conference on Pervasive Computing and Communications Workshops and other Affiliated Events (PerCom Workshops), Kassel, Germany, 22-26 Mar. 2021; pp. 680-685. DOI: 10.1109/PerComWorkshops51409.2021.9431086.

[16] Y. Jia; F. Zhong; A. Alrawais; B. Gong; X. Cheng. FlowGuard: an intelligent edge defense mechanism against IoT DDoS attacks. IEEE Internet of Things Journal 2020, 7, 10, pp. 9552–9562. DOI: 10.1109/JIOT.2020.2993782

[17] J. Su; D. V. Vasconcellos; S. Prasad; D. Sgandurra; Y. Feng; K. Sakurai. Lightweight classification of IoT malware based on image recognition. In Proceedings of 2018 IEEE 42nd Annual Computer Software and Applications Conference (COMPSAC), Tokyo, Japan, 23-27 Jul. 2018; 2, pp. 664-669. DOI: 10.1109/COMPSAC.2018.10315

Point 6: The manuscript requires English language grammar editing. Several sentences are hard to comprehend fully. The Author(s) is advised to reach out to editors or use editing tools for corrections regarding the use of language.

Response 6: Thank you very much for this comment. The English writing has been checked carefully to be improved in the revised version.

Point 7: Abbreviations used in the study should be given in their long-form only where they are first used. There is no need to use the long forms again later. For example, the Internet of Things and denial of service, convolutional neural network, etc.

Response 7: Thank you very much for this comment. The comment has been followed in the revised version.

Point 8: The advantages and disadvantages of the study should be explained in detail in order to guide the reader.

Response 8: Thank you very much for this comment. The advantages and disadvantages of the study are described below and have been added in Section 5.

“This DDoS detection system can be built properly without changing the original IoT hardware structure due to adopting the edge computing architecture. In addition, two 2D CNN models trained by packet traffic and packet features data respectively are used simultaneously to identify DDoS attacks that can effectively improve the accuracy of identification. However, the 2D CNN models only trained for usual DDoS attacks will cause reducing the accuracy of identification for new types of DDoS attacks. Moreover, the datasets used in the experiments in this paper were completely captured from the experimental network which will result in the result of the paper falling into a special case. Thus, using an open dataset to verify this system is necessary for the future.”

Point 9: As such, the study presents a small simulation for the detection of distributed denial of service attacks. The study does not contribute enough to the scientific literature. For example, it is not clear how much bandwidth the work will be sufficient for the attack that can be made. Again, it is not clear from the model point of view how many nodes or how many simultaneous attacks the study can detect. It is not clear that the work as such would protect a real system or how much bandwidth it would protect. The original value of the study is unfortunately weak due to the deficiencies it contains. I do not think this version of the work is acceptable.

Response 9: Appreciate your comments. We have carefully checked the manuscripts, and added more materials to respond the queries of the reviewers. This paper completed the two 2D CNN models trained by packet traffic and packet features data respectively are used simultaneously to identify DDoS attacks that can effectively improve the accuracy of identification. In this work, we build an AIoT platform for the experiments to identify DDoS attacks with WiFi Networks. Therefore, in the experimental environments of platform, the bandwidth and how many nodes will depend on which type of the applied WiFi Networks. Moreover, in this work, we have referred the updated reference of [8-10] to further perform real experiments in which two 2D CNN models trained by packet traffic and packet features data respectively are used simultaneously to identify DDoS attacks that can effectively improve the accuracy of identification.

Reviewer 2 Report

In this paper, a DDoS attack detection system based on edge computing on an IoT system is proposed. The 2D CNN model is trained to identify whether the data collection server in the Internet of Things is under a DDoS attack. The detailed comments can be summarised as follows:

Do any existing studies simulate DDoS attacks using CNN on IoT systems? It is unclear why the authors did not compare the results with those in Refs [17] and [18].

Therefore, the existing studies have not been addressed, and the research gap remains unclear.

There is no justification for the results, especially the figures.

There has not been a discussion of the rationale behind the methodology used.

What is the reason for not considering the confusion matrix in the evaluation?

Author Response

Response to Reviewer 2 Comments

In this paper, a DDoS attack detection system based on edge computing on an IoT system is proposed. The 2D CNN model is trained to identify whether the data collection server in the Internet of Things is under a DDoS attack. The detailed comments can be summarised as follows:

Point 1: Do any existing studies simulate DDoS attacks using CNN on IoT systems? It is unclear why the authors did not compare the results with those in Refs [17] and [18]. Therefore, the existing studies have not been addressed, and the research gap remains unclear.

Response 1: Thank you very much for this comment. This work focuses on creating a real experiment IoT system to evaluate if the proposed DDoS attacks detection system performs well or not. The datasets used for training machine learning models are captured from the uniquely created IoT system. Therefore, the results of this study cannot be compared properly to the ones of other literature due to the different research platforms. However, we could adopt total software simulations on an objective platform to evaluate the proposed system and compare it to existing studies in the future.

Point 2: There is no justification for the results, especially the figures.

Response 2: Thank you very much for this comment.

A more detailed explanation for Figures 16 and 17 has been made in the revised version in Section 4.

“Figures 16 (a)-(e) show the total number of TCP, UDP, ICMP, and other packets under the normal transmission, SYN flood attack, UDP flood attack, ICMP flood attack, and MIX flood attack, respectively. From Figure 16(a), it can be seen that under normal transmission conditions, there are all kinds of packets, and the cumulative number of packets increases moderately. However, no matter what kind of flood attack, the cumulative number of packets increases suddenly. Moreover, there are different cumulative numbers for different packet types when under attack. These characters can not only be used to judge whether an attack has been suffered, but also can be used to identify the type of attack.

The experimental retrieval time is 5 minutes, so there are 100 records. Compared with Figure 16(a), it can be seen in Figure 16(b) that suffered from SYN flood attack, its TCP packets increased a lot, and the total number of TCP packets flowing through IP1 within 5 minutes was about 50,000. And other types of packets are almost equal to 0. In Figure 16(c) of the UDP flood attack, it can be seen that UDP packets have increased a lot, while other types of packets are almost equal to 0. The collection time is 5 minutes. It can be seen that the sum of UDP packets during this period has reached about 40000. In Figure 16(d) of the ICMP flood attack, it can be seen that ICMP packets have increased a lot, while other types of packets are almost equal to 0, and the collection time is 5 minutes. During this period of time, the sum of ICMP packets also reached about 40,000. Figure 16(e) shows the situation of being attacked by other packets (MIX flood attack). At this time, the server flows through a large number of various types of packets, and the number of packets with UDP and TCP characteristics is almost the same. Therefore, in our experiment, we can It is obvious to observe that when attacked, the increase of various packets with this time, and the change of the number of packets.

Figures 17 (a)-(e) are the usage rates of their CPU and memory, respectively. As can be seen from Figure 17 (a), the CPU usage rate changes little during normal transmission, but it will fluctuate greatly under attacks. Similarly, the memory usage rate is much lower under normal transmission than under attacks. It helps to detect whether the system is under attack. Table 4 shows the average results of ten different speed tests conducted by iPerf in AP1 for the above five cases with a transmission file size of 352Kb.

In Figure 17(b), it can be seen that when a SYN flood attack is encountered, the usage rate of CPU is about 20% higher than that in Figure 17(a) under normal conditions. As shown in Figure 17(c), when suffering from UDP flood attacks, the CPU usage rate fluctuates violently, and its highest usage rate reaches more than 50%. In Figure 17(d), we can see that when the ICMP flood attack is encountered, the CPU’s usage rate fluctuates violently. The situation of the oscillation is the same as that of the UDP flood attack, but it has different characteristics. In Figure 17(e), it can be seen that when the MIX flood is encountered During the attack, the CPU’s usage rate will suddenly rise and fluctuate violently, just like the previous attacks. As shown in 17(b)-(e), the usage rate of the memory remains roughly the same as normal. Therefore, we can find out its characteristics from the fluctuation of CPU usage. From the results in Table 4, it can be seen that various attacks will affect the network speed during transmission. In our experiment, the average network transmission speed will drop from an average of 14.2 Mbps to about 0.3 Mbps. Based on the above experimental results, we use neural network for edge computing and model training.

Point 3: There has not been a discussion of the rationale behind the methodology used.

Response 3: Thank you very much for this comment. The rationale of the proposed method has been added in the revised version. Please refer to Introduction Section.

“ However, the identification results on the edge computer when measured under different conditions, in which Feature is the identification result of the feature model, Flow means the identification result of the traffic model. Therefore, we conduct our research using the related neural network approach. The proposed architecture of edge computing with a trained CNN model will make correct identifications under the normal transmission, SYN flood attack, UDP flood attack, ICMP flood attack, and MIX flood attack respectively. In this proposal, we would like to maximize the use of information about changes in related characteristics when packets are transmitted in the system, which can effectively improve the accuracy of recognition and make up for the shortcomings of using a single model for judgment.

 Moreover, the discussion of rationale of the proposed method has been described in Section 4.

“Table 8 lists the identification results on the edge computer when measured under different conditions, in which Feature is the identification result of the feature model, Flow means the identification result of the traffic model, and Weighted is the identification result after individual weighting on the two models. According to the table, the proposed architecture of edge computing with a trained CNN model can make correct identifications under the normal transmission, SYN flood attack, UDP flood attack, ICMP flood attack, and MIX flood attack respectively. Even in the lack of packets case during normal trans-mission, the feature model cannot recognize it normally, but the traffic model can still recognize it when the data is rare. Since the traffic model accounts for 70% of the weight, the weighted identification result is normal. The above experimental results are mainly because this system adopts two independent 2D CNN models trained by the packet traffic dataset and packet features dataset respectively, and recognizes the existing situation of the system at the same time, and then adds appropriate weights to them as a basis for judgment. This design can maximize the use of information about changes in related characteristics when packets are transmitted in the system, which can effectively improve the accuracy of recognition and make up for the shortcomings of using a single model for judgment.

Point 4: What is the reason for not considering the confusion matrix in the evaluation?

Response 4: Thank you very much for this comment. Because the models established by the data in our experimental environment have good TP and TN values in the end. So, there is no consideration the confusion matrix. But, it is good advice enlightening us considering the confusion matrix in future further study.

Reviewer 3 Report

The author wrote the paper in one flow ,covering all required background. The application of Internet of thing is well explained and challenges were highlighted. The issue related to information security is due to distributed denial of service attack. The environment is created in real data collection server and propose an autonomous defence system. The scenario is well narrated and DDoS attack at different level were considered  and edge computing technique is used for data collection. The flow chart explains the algorithm from transmitter uptu final reception of data in IOT environment The model basically consider four different attack environment. They are SYN flood attack, UDP flood attack ICMP flood attack and MIX flood attack. The 2D CNN trained model was used for verification. Overall paper has novelty and  weightage concept is used. As a future scope some more attacks can be taken intu consideration and its impact can be analyzed with different weight.

Author Response

Response to Reviewer 3 Comments

Point 1: The author wrote the paper in one flow, covering all required background. The application of Internet of thing is well explained and challenges were highlighted. The issue related to information security is due to distributed denial of service attack. The environment is created in real data collection server and propose an autonomous defence system.

Response 1: Thank you very much for this comment.

Point 2: The scenario is well narrated and DDoS attack at different level were considered and edge computing technique is used for data collection. The flow chart explains the algorithm from transmitter uptu final reception of data in IOT environment.

Response 2: Thank you very much for this comment.

Point 3: The model basically consider four different attack environment. They are SYN flood attack, UDP flood attack ICMP flood attack and MIX flood attack. The 2D CNN trained model was used for verification. Overall paper has novelty and weightage concept is used.

Response 3: Thank you very much for this comment.

Point 4: As a future scope some more attacks can be taken into consideration and its impact can be analyzed with different weight.

Response 4: Thank you very much for this comment. Thank you very much for this comment. It is good advice enlightening us considering its impact with different weight in future study.

Round 2

Reviewer 1 Report

The paper suggested deep learning-based techniques for detecting and preventing numerous forms of distributed denial-of-service (DDoS) attacks on two generated datasets using internet of things (IoT) devices integrated with edge computing. By employing Raspberry pi, DHT11 sensor, and edge computing approach, packet traffic and packet feature datasets were created via the use of two IP addresses by the constructed IoT system. After appropriate pre-processing, a neural network, 1D, and 2D CNN models were used to detect SYN, UDP, ICMP, and MIX flood DDoS attacks. The result from the 2D CNN model proved to be better. 

The authors were ordered to make several changes due to the existence of numerous corrections in the initial proposed study. After making the necessary changes and reviews, it is suggested that it can be considered for publication. 

Thank you.